# Biosensor Based on Covalent Organic Framework Immobilized Acetylcholinesterase for Ratiometric Detection of Carbaryl

**DOI:** 10.3390/bios12080625

**Published:** 2022-08-10

**Authors:** Ying Luo, Na Wu, Linyu Wang, Yonghai Song, Yan Du, Guangran Ma

**Affiliations:** National Engineering Research Center for Carbohydrate Synthesis, Key Lab of Fluorine and Silicon for Energy Materials and Chemistry of Ministry of Education, College of Chemistry and Chemical Engineering, Jiangxi Normal University, Nanchang 330022, China

**Keywords:** carbaryl, acetylcholinesterase, covalent organic framework, inhibition-based electrochemical biosensor

## Abstract

A ratiometric electrochemical biosensor based on a covalent organic framework (COF_Thi-TFPB_) loaded with acetylcholinesterase (AChE) was developed. First, an electroactive COF_Thi-TFPB_ with a two-dimensional sheet structure, positive charge and a pair of inert redox peaks was synthesized via a dehydration condensation reaction between positively charged thionine (Thi) and 1,3,5-triformylphenylbenzene (TFPB). The immobilization of AChE on the positively charged electrode surface was beneficial for maintaining its bioactivity and achieving the best catalytic effect; therefore, the positively charged COF_Thi-TFPB_ was an appropriate support material for AChE. Furthermore, the COF_Thi-TFPB_ provided a stable internal reference signal for the constructed AChE inhibition-based electrochemical biosensor to eliminate various effects which were unrelated to the detection of carbaryl. The sensor had a linear range of 2.2–60 μM with a detection limit of 0.22 μM, and exhibited satisfactory reproducibility, stability and anti-interference ability for the detection of carbaryl. This work offers a possibility for the application of COF-based materials in the detection of low-level pesticide residues.

## 1. Introduction

Carbaryl as a kind of carbamate pesticide has been widely applied in agricultural products due to their short-term toxicity and high insecticidal activity [1,2,3,4]. However, because of the bioaccumulation effect, a large amount of residual carbaryl in water, soil, food and the environment enter the human body through the skin, respiratory tract, digestive tract, etc., resulting in irreversible damage [5,6,7,8]. Therefore, it is crucial to achieve rapid detection and reliable quantification of carbaryl. Traditional detection methods, such as chromatography [9,10,11,12], surface-enhanced Raman spectroscopy [13,14,15], immunoassay [16,17], etc., have been developed very well, with high sensitivity and accuracy in the determination of pesticide residues in water and agricultural products. However, the limitations lie in the complex sample handling process, the use of highly toxic organic solvents and the expensive and complex instruments that require professional testing [18,19,20].

Nowadays, acetylcholinesterase (AChE) inhibition-based electrochemical biosensors have attracted great attention with regard to the detection of carbaryl and other carbamate pesticides due to their advantages of non-toxicity, simplicity, miniaturized, high specificity and high sensitivity [21,22,23]. For example, Loguercio et al. established the biosensor for the detection of carbaryl by immobilizing AChE on the polypyrrole nanocomposite, showing satisfactory results [24]. Zhang et al. used graphene as the support material to load AChE, and the constructed sensor realized the chiral recognition of (+)/(−)-methamidophos [25]. Considering that acetic acid, the hydrolysate of acetylthiocholine (ATCh), can induce the collapse of unstable metal-organic frameworks (MOFs), Li et al. prepared a biodegradable ZIF-8/MB composite using the one-pot method and realized ultrasensitive detection of paraoxon by using AChE as the recognition molecule [26]. The mechanism of the AChE inhibition-based electrochemical biosensors is that the carbamate pesticides make the serine residue hydroxyl group of the AChE catalytic center be carbamylated, resulting in a decrease in its activity or even making it completely inactive. Thus, the catalytic hydrolysis of ATCh is weakened, leading to a decrease in the production of thiocholine (TCh). The response current of TCh is inversely proportional to the concentration of carbamate pesticides such as carbaryl; therefore, this causes the simple, efficient and sensitive detection of carbaryl [27,28,29,30,31].

The following two points are of great significance for the construction of high-performance AChE inhibition-based pesticide electrochemical sensors. Firstly, choose an appropriate support material to immobilize the enzyme and maintain its activity [32,33,34]. Suitable electrode support materials should allow a large number of enzymes to be loaded and provide a good microenvironment for maintaining enzyme activity [35,36,37]. Yang et al. reported that fixing AChE on the support material with a positive charge not only favors the maintenance of its bioactivity, but also promotes electron transport between the AChE and electrode surface [38]. Secondly, the influence of the background current and changeable environmental conditions on sensor performance should be avoided [39,40,41]. The traditional single-signal electrochemical sensors have low accuracy and sensitivity and poor reproducibility because their electrochemical signal is easily affected by the background current of the workstation and environmental conditions such as temperature and pH [42,43]. Fortunately, dual-signal ratiometric electrochemical sensors have emerged [44,45,46]. Wang et al. coated an electroactive covalent organic framework (COF_Thi-TFPB_) on the surface of carbon nanotubes (CNTs) using the one-pot method, and the prepared COF_Thi-TFPB_-CNT nanocomposite was used for electrochemical ratiometric detection of AA. Since the monomer thionine (Thi) was positively charged, the positively charged COF_Thi-TFPB_ could self-peel into large-sized two-dimensional crystal nanosheets, and a pair of redox peaks of the COF_Thi-TFPB_ was inert to the detection of AA; thus, it could be used as a reference signal [47].

Here, an electrochemical sensor based on the COF_Thi-TFPB_ [48,49,50,51] loaded with AChE for the detection of carbaryl is proposed. The positively charged COF_Thi-TFPB_ can be easily stripped into two-dimensional nanosheets, and modified on the surface of a bare glassy carbon electrode (GCE) without adhesives or conductive agents. The positively charged COFThi−TFPB is conducive to effectively maintaining the bioactivity of AChE and achieving the best catalytic effect. Its inherent redox peak at 0/−0.22 V is inert to the detection of carbaryl, which could be used as a reference signal to further improve the sensitivity and accuracy of the detection. The prepared sensor shows good reproducibility, stability and anti-interference ability. This work proposes an efficient strategy to immobilize enzymes using an electroactive COF as a support material.

## 2. Experimental Procedure

### 2.1. Materials and Reagents

1,3,5-triformylphenylbenzene (TFPB) and thionine (Thi) were purchased from Jilin Yanshen Technology Co., Ltd., (Beijing, China). N,N-dimethylformamide (DMF), N,N-dimethylacetamide (DMA), mesitylene, tetrahydrofuran (THF), acetic acid (AcOH), acetylcholinesterase (AChE), acetylthiocholine (ATCh) and other chemicals were purchased from Inokay Co., Ltd. (Beijing, China). Carbaryl was purchased from Mokai Nike Technology Co., Ltd. (Jiangxi, China). 

### 2.2. Instruments

Scanning/transmission electron microscopy images (SEM/TEM) were obtained via the HITACHI S-3400N SEM and JM-2010 (HR) TEM (Chiyoda City, Japan), respectively. Atomic force microscopy (AFM) images were obtained via the instrument model BRUKER Nanoscope V (MultiMode 8) (Billerica, MA, USA) multifunctional scanning probe microscope. Fourier transform infrared spectroscopy (FTIR) was recorded on model Perkin-Elmer Spectrum 100 spectrometer (Waltham, MA, USA). N_2_ adsorption/desorption isotherm measurements were operated using a BELSORP-mini II instrument (Microtrac, Haan/Duesseldorf, Germany) under the liquid nitrogen temperature of 77 K. Powder X-ray diffraction (XRD) analysis was performed on the D/Max 2500 V/PC X-ray powder diffractometer (Rigaku, Tokyo, Japan) with a scanning step of 1°/min. All electrochemical studies were performed on an electrochemical workstation (CHI 760D, Shanghai, China).

### 2.3. Preparation of COF_Thi-TFPB_

Firstly, 0.2 mM TFPB and 0.3 mM Thi were added to a mixture with 2 mL of 1,4-dioxane, and 1 mL of mesitylene and DMF, and ultrasound was performed for 15 min. Next, it was transferred to a 25 mL reaction kettle with 0.2 mL (concentration: 6 M) acetic acid (used as an initiator), and placed in an oven at 120 °C for three days. Finally, the dark-blue COF_Thi-TFPB_ was obtained by centrifugation and freeze-drying [52].

### 2.4. Preparation of AChE/COF_Thi-TFPB_/GCE

Firstly, the surface of the glassy carbon electrode (GCE) was treated with Al_2_O_3_, ethanol and ultrapure water until smooth and clean. Then, 5 μL of the 2 mg/mL COF_Thi-TFPB_ was dropped on the electrode surface. After drying, AChE/COF_Thi-TFPB_/GCE was prepared by dropping 0.4 mM AChE on the modified electrode. The detection mechanism of carbaryl is shown in Figure 1.

## 3. Results and Discussion

### 3.1. Characterization of COF_Thi-TFPB_

SEM (Figure 1a), TEM (Figure 1b) and AFM (Figure 1c) showed that the COF_Thi-TFPB_ owned a film-like lamellar structure, and the thickness was about 1.42 nm, which was very favorable for the immobilization of AChE [53,54]. Next, an FTIR spectrum and XRD pattern were used to demonstrate the successful synthesis of the COF_Thi-TFPB_. The spectrum of the COF_Thi-TFPB_ showed that a new peak of -C=N- appeared at 1658 cm^−1^, whereas the disappearance of N−H in –NH_2_ at 3292 cm^−1^ and C=O in –CHO at 1689 cm^−1^. Meanwhile, the stretching vibration peaks corresponding to –CH in –CHO at 2714 cm^−1^ and 2834 cm^−1^ disappeared. These results demonstrated that the COF_Thi-TFPB_ was synthesized successfully (Figure 1d). The XRD pattern further confirmed the successful synthesis of the crystalline COF_Thi-TFPB_ (Figure 1e). Diffraction peaks at 6.67°, 8.48°, 11.34°, 25.8° and 45.6° corresponded to (100), (110), (210), (152) and (111) crystal planes. The N_2_ adsorption/desorption isotherm of the COF_Thi-TFPB_ showed that the specific surface area was 67.4 m^2^ g^−1^ (Figure 1f).

### 3.2. Electrochemical Behaviors of COF_Thi-TFPB_/GCE and AChE/COF_Thi-TFPB_/GCE

To successfully construct an electrochemical sensor for ratiometric detection of carbaryl, the introduced internal reference signal should have good stability and a suitable oxidation potential. Therefore, the electrochemical performance of the COF_Thi-TFPB_ was evaluated by cyclic voltammetry (CV). As shown in Figure 2a, compared with the TFPB/GCE (b) without a peak, both the COF_Thi-TFPB_/GCE (c) and Thi/GCE (a) had two pairs of redox peaks at the same potential, indicating that the peaks on the COF_Thi-TFPB_/GCE came from the electroactive Thi. Next, the CV of the COF_Thi−TFPB_/GCE was investigated at different scan rates. It could be seen that the positions of their redox peaks were basically unchanged with the increase in scanning rate (Figure 2b). The redox peak at −0.2/−0.07 V was caused by the electrochemical reaction of Thi in the COF_Thi-TFPB_, and the redox peak at 0/−0.22 V was due to the conjugated structure of Thi in the COF_Thi-TFPB_ [55]. With the increase in scanning rate, the peak current density of the two redox peaks of the COF_Thi-TFPB_ also increased and showed a good linear relationship, indicating that the reaction process was a typical surface control process (Figure 2c). Based on the linear regression equation between the anodic/cathodic peak potential and natural logarithm of the scan rate (Figure 2d), it could be calculated that the electron-transfer number (n) was 1, and the electron-transfer coefficient (αs) was 0.369 when the redox potential was 0/−0.22 V [56]. 

Then, CV and electrochemical impedance spectroscopy (EIS) tests were performed using 5.0 mM [Fe(CN)_6_]^3−/4−^ in a 0.1 M KCl solution as a probe to investigate the electrochemical behaviors of the COF_Thi-TFPB_/GCE and AChE/COF_Thi-TFPB_/GCE. As shown in Figure 3a, compared with the bare GCE (curve a), the peak current of [Fe(CN)_6_]^3−/4−^ on the COF_Thi-TFPB_/GCE (curve b) was slightly increased, and the peak-to-peak potential difference was slightly decreased. This result might be attributed to the electrostatic attraction between the negative [Fe(CN)_6_]^3−/4−^ and the positive COF_Thi-TFPB_. In the meantime, the redox peak of [Fe(CN)_6_]^3−/4−^ on the AChE/COF_Thi-TFPB_/GCE (curve c) owned the smallest peak current and the largest peak-to-peak potential difference. It was mainly due to the poor conductivity of AChE, which would hinder the electron transport between [Fe(CN)_6_]^3−/4−^ and the surface of the electrode [57,58]. The charge transfer resistance (*R*_ct_) values of the bare GCE, COF_Thi-TFPB_/GCE and AChE/COF_Thi-TFPB_/GCE were 31.3 Ω, 8.5 Ω and 288.1 Ω, respectively (Figure 3b). In conclusion, compared with the COF_Thi-TFPB_/GCE, the peak current of [Fe(CN)_6_]^3−/4−^ on the AChE/COF_Thi-TFPB_/GCE decreased and the *R*_ct_ value increased, which directly proved the successful fixation of AChE on the COF_Thi-TFPB_/GCE. 

### 3.3. Optimization of the Experimental Conditions

It was known that when at the optimum pH value, the binding ability of the enzyme molecule to the substrate was the strongest and the enzyme reaction rate was the highest; however, if the pH was too large or too small, the enzyme might be inactivated [59,60]. Therefore, the pH value of the solution was optimized. As shown in Figure 4a, when pH = 7.0, the current density of the AChE/COF_Thi-TFPB_/GCE was the largest in the 0.1 M PBS with 0.6 mM ATCh and 5 μM carbaryl. Then, the amount of the COF_Thi-TFPB_ modified on the electrode surface was optimized (Figure 4b), which showed that the optimal volume was 5 μL (concentration: 2 mg/mL). Next, considering that the amount of loaded AChE and the concentration of substrate molecule ATCh had important influence on the detection results, they were also optimized. It could be observed that it was best to set the concentration of AChE and ATCh at 0.4 mM and 0.6 mM in the subsequent experiments, respectively (Figure 5a,b). Figure 5c showed the relationship between the catalytic activity inhibition of AChE and the incubation time. The inset is the formula for calculating the percentage of inhibition (*I*%), where *j*_P,control_ was the original current density recorded by the AChE/COF_Thi-TFPB_/GCE in 0.1 M PBS (pH = 7.0) with 0.6 mM ATCh, *j*_P,exp_ was the residual current density recorded after immersing in 0.1 M PBS (pH = 7.0) with 0.6 mM ATCh and 5 μM carbaryl for 0, 4, 8, 12, 16, 20 and 30 min. All in all, when the concentration of AChE was 0.4 mM, ATCh was 0.6 mM and the incubation time was 20 min, the performance of the AChE/COF_Thi-TFPB_/GCE sensor was the best.

### 3.4. Electrochemical Detection of Carbaryl Based on AChE/COF_Thi-TFPB_/GCE

Firstly, the affinity of AChE fixed on the COF_Thi-TFPB_ to ATCh was investigated. Figure 6a shows the relation curve between response current density and time after continuously adding ATCh. It could be seen that there was a good linear relationship between the oxidation peak current density and the concentration of ATCh between 0.01 mM and 0.27 mM. However, the slow-response current density at higher concentrations of ATCh indicated a Michaelis–Menten process. According to the slope and intercept of the linear regression equation in Figure 6b, the Michaelis–Menten constant (Km) was calculated to be 0.24 mM. This value was lower than 0.622 mM as measured by the AChE/COF@MWCNTs/GCE [61], suggesting good affinity between the enzyme and the substrate. Then, a ratiometric electrochemical sensor based on the AChE/COF_Thi-TFPB_ was used to detect carbaryl in 0.1 M PBS (pH = 7.0) containing 0.6 mM ATCh. As shown in Figure 7a, the peak current density of the COF_Thi-TFPB_ at -0.05 V was basically unchanged with the addition of carbaryl, whereas the peak current density of TCh at 0.6 V gradually decreased. This was because the toxic effect of carbaryl on AChE led to a decrease in the amount of TCh, and then, the electrochemical signal was weakened. The inset in Figure 7a shows the linear relationship between *j*_TCh_/*j*_COF_ and the concentration of carbaryl, where the linear range of the carbaryl sensor was 2.2–60 μM and the detection limit was 0.22 μM. The performance of the AChE/COF_Thi-TFPB_/GCE sensor was compared with other sensors (Table 1). It could be seen that the detection limit of this sensor was lower than that based on Au/PAMAM/GLUT/AChE (3.2 μM) and MWCNT/PANI/AChE (1.4 μM).

The selectivity of the AChE/COF_Thi-TFPB_/GCE sensor was investigated in 0.1 M PBS (pH = 7.0) with 0.6 mM ATCh, 10 μM carbaryl and 50 μM interferences. It could be seen that these interferences had little effect on the peak current density (Figure 7b). Then, one AChE/COF_Thi-TFPB_/GCE was used to measure the corresponding peak current density of 5 μM carbaryl in 0.1 M PBS (pH = 7.0) with 0.6 mM ATCh for 30 days. The relative standard deviation (RSD) was only 1.15%, indicating that the AChE/COF_Thi-TFPB_/GCE sensor had good stability (Figure 7c). The RSD of 5 μM carbaryl detected by six independent AChE/COF_Thi-TFPB_/GCEs was 1.45% in 0.1 M PBS (pH = 7.0) with 0.6 mM ATCh (Figure 7d). The good stability and reproducibility might be attributed to the fact that the positively charged COF_Thi-TFPB_ could immobilize the AChE enzyme and maintain its activity, and its oxidation peak played a self-correcting role in the detection of carbaryl.

### 3.5. Detection of Carbaryl in Vegetable Samples

In addition, the AChE/COF_Thi-TFPB_/GCE sensor and high-performance liquid chromatography (HPLC) were used to detect carbaryl in real samples to demonstrate the practical application capability of the sensor. Firstly, a 100 g lettuce sample was chopped and put into a juicer containing 100 mL of 0.1 M PBS (pH = 7.0). Then, the obtained mixture was filtered, and the filtrate was used as the actual sample. Next, different concentrations of carbaryl were added to the actual sample, and the carbaryls in the actual samples were determined by the AChE/COF_Thi-TFPB_/GCE sensor and HPLC. The obtained results are shown in Table 2. It could be seen that the carbaryl content in the actual samples obtained by the AChE/COF_Thi-TFPB_/GCE sensor was close to the results of the HPLC test, which proved that the AChE/COF_Thi-TFPB_/GCE sensor has the potential to detect carbaryl in real examples.

## 4. Conclusions

The development of efficient, good, stable and reproducible AChE inhibition-based electrochemical biosensors might rely on the immobilization of the enzyme on suitable support materials and the introduction of internal reference signals to eliminate irrelevant effects in detection. In this work, a ratiometric electrochemical sensor was constructed by using the positively charged COF_Thi-TFPB_ with an inert redox peak as the support material to load the AChE enzyme. The COF_Thi-TFPB_ could immobilize the AChE enzyme and maintain its activity. On the other hand, its inherent redox peak at 0/−0.22 V was inert to the detection of carbaryl, which could be used as a reference signal to further improve the sensitivity and accuracy of the detection. The linear range of this sensor was 2.2–60 μM, the detection limit was 0.22 μM, and it had good selectivity, reproducibility and stability. This suggests that this material has the potential to be applied to detect low-level pesticide residues.

## Data Availability

The data is available under the request to the correspondence.

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
