# Peer review of "Biosensor Based on Covalent Organic Framework Immobilized Acetylcholinesterase for Ratiometric Detection of Carbaryl"

_biosensors, 2022, doi:10.3390/bios12080625_

Round 1

Reviewer 1 Report

Manuscript ID: Biosensors-1846167

Title: A novel acetylcholinesterase biosensor based on electroactive organic framework for ratiometric detection of carbaryl

Authors: Ying Luo, Na Wu, Linyu Wang, Yonghai Song, Yan Du and Guangran Ma*

This article describes the COFThi−TFPB with acetylcholinesterase for carbaryl sensing.

There are some major problems in the manuscript and a careful review is needed.

1. The title need to be modified and more specific. Why do the authors mention acetylcholinesterase biosensor, are the sensors sensing the carbaryl or acetylcholinesterase?  

2. What is the novelty in the manuscript? The nanomaterial, COFThi−TFPB was reported previously in references 45 and 50. The authors dropped another layer of a known enzyme, acetylcholinesterase.

3. The author claims COFThi−TFPB itself electroactive material. Therefore, they can study the electrochemical behavior of COFThi−TFPB without an external redox probe, Fe(CN)63−/4−. Similar to reference 45, figure 5A.

4. In Table 1, the detection limit of other sensors is better than this work except for two references, and also the incubation time was 20 minutes. how this sensor would be suitable for carbaryl detection in pesticides?

5. The authors can study the apparent Michaelis–Menten constant of the immobilized enzyme. Similar to reference 60, figure 5.

For all these reasons I do not recommend the publication of this article in the biosensors journal in the present form.

Author Response

Thank you very much for giving us very valuable suggestions. We have considered your questions seriously, and answered them as below.

Reviewer 1: This article describes the COFThi−TFPB with acetylcholinesterase for carbaryl sensing. There are some major problems in the manuscript and a careful review is needed.

1.Q: The title need to be modified and more specific. Why do the authors mention acetylcholinesterase biosensor, are the sensors sensing the carbaryl or acetylcholinesterase?

R: Thank you very much for your relevant advice. According to your advice, we have changed the title to “Biosensor based on covalent organic framework immobilized acetylcholinesterase for ratiometric detection of carbaryl”. (See line 1-3 on page 1 in the revised manuscript).

2.Q: What is the novelty in the manuscript? The nanomaterial, COFThi−TFPB was reported previously in references 45 and 50. The authors dropped another layer of a known enzyme, acetylcholinesterase.

R: Thank you very much for your relevant advice. In reference 45, COFThiTFPB-CNT composites were prepared for electrochemical sensing of ascorbic acid (AA) and pH. Here CNTs could well catalyze the oxidation of AA, while COFThiTFPB had a pair of redox peaks that were inert for AA as the reference signal. In reference 45, the COFTFPB-Thi was grown vertically on three-dimensional porous carbon derived from kenaf stem (3D-KSC) for double signal ratiometric electrochemical detection of RF. The COFTFPB-Thi showed two reduction peaks at -0.08 V and -0.23 V, which came from the reduction of COFTFPB-Thi and the conjugated structure of COFTFPB-Thi, respectively. In the presence of RF, those RF molecules near the electrode surface were oxidized at 0.6 V. Then some oxidized RF (RFox) adsorbed on COFTFPB-Thi would oxidize COFTFPB-Thi into COFTFPB-Thi(ox) while other RFox adsorbed on 3D-KSC kept unchanged. When the potential was scanned from 0.6 V to -0.6 V, both COFTFPB-Thi(ox) and RFox adsorbed on 3D-KSC were reduced at -0.08 V and -0.45 V accordingly, while the reduction peak of -0.23 V of the conjugated structure of COFTFPB-Thi kept constant. When j-0.45/j-0.23 was used as the response signal, the detection limit was 44 nM and the linear range was 0.13 μM -0.23 mM. By using j-0.08/j-0.23 as the response signal, a detection limit of 90 nM and a linear range of 0.30 μM-0.23 mM (S/N=3) were obtained. By using double signals, the measurement results can be corrected to make the results more accurate and reliable. It is well known that how to immobilize the enzyme on the electrode surface to maximize the bioactivity of the enzyme is very important for biosensors. In our work, the COFTFPB-Thi was used to load AChE for the first time. The positively charged COFThi−TFPB is conducive to effectively maintaining the biological activity of AChE and achieving the best catalytic effect, which is very important for constructing biosensor. Furthermore, its inherent redox peak at 0/-0.22 V is inert to the detection of carbaryl, which could be used as a reference signal to further improve the sensitivity and accuracy of the detection. The corresponding discussion has been added in the revised manuscript. (See the revised manuscript)

3.Q: The author claims COFThi−TFPB itself electroactive material. Therefore, they can study the electrochemical behavior of COFThi−TFPB without an external redox probe, Fe(CN)63−/4−. Similar to reference 45, figure 5A.

R: Thank you very much for your relevant advice. According to your suggestion, we have investigated the electrochemical behavior of COFThi−TFPB without an external redox probe, Fe(CN)63−/4− and added the following contents “As shown in Fig. 2a, compared with TFPB/GCE (b) without a peak, both COFThi-TFPB/GCE (c) and Thi/GCE (a) had a pair of redox peaks at the same potential at about -0.2 V, indicating that the peaks on COFThi-TFPB/GCE came from the electroactive center Thi.” in the revised manuscript. (See line 10-14 on page 7 in the revised manuscript). (See line 1 on page 27 in the revised manuscript).

4.Q: In Table 1, the detection limit of other sensors is better than this work except for two references, and also the incubation time was 20 minutes. How this sensor would be suitable for carbaryl detection in pesticides?

R: Thank you very much for your relevant advice. In fact, the detection limit of many sensors is poorer than that our works as shown in revised Table 1. Furthermore, our sensor has the following advantages as compared to other sensors. On the one hand, the introduction of the inert redox peaks of COFThi-TFPB can ensure that the response signal and reference signal for the detection of carbaryl are mutually dependent in the same sensing environment, so as to achieve internal correction and improving the accuracy of detection. The positively charged COFThi−TFPB was conducive to maintaining the biological activity of AChE and achieving the best catalytic effect, and the lower Michaelis-Menten Constant (Km) indicates that AChE has a good affinity with ATCh. The corresponding discussion has been added in the revised manuscript. (See line 1 on page 33 in the revised manuscript)

5.Q: The authors can study the apparent Michaelis–Menten constant of the immobilized enzyme. Similar to reference 60, figure 5.

R: Thank you very much for your relevant advice. According to your suggestion, we have investigated the affinity of AChE immobilized on COFThi-TFPB for the substrate molecule ATCh and added the following contents. “Firstly, the affinity of AChE fixed on COFThi-TFPB to ATCh was investigated. Fig. 6a showed the relation curve between response current density and time after continuously adding ATCh. It could be seen that there was a good linear relationship between the oxidation peak current density and the concentration of ATCh between 0.01 mM and 0.27 mM. However, the slow response current density at higher concentrations of ATCh was suggestive of a Michaelis-Menten process. According to the slope and intercept of the linear regression equation in Fig. 6b, the Michaelis-Menten Constant (Km) was calculated to be 0.24 mM. This value was lower than 0.622 mM measured by AChE/COF@MWCNTs/GCE [61], suggesting good affinity between the enzyme and the substrate.” in the revised manuscript. (See line 16-23 on page 9 and line 1 on page 10 in the revised manuscript).

Reviewer 2 Report

This manuscript constructed an electrochemical sensing platform for the quantitative monitoring of carbaryl. The ratiometric electrochemical sensor was fabricated by using acetylcholinesterase-loaded electroactive covalent organic framework. The electroactive COF was modified with thionine. Results indicated that the developed electrochemical sensing system could exhibit good analytical properties. However, modification is requested for the further consideration as follows:

1.      Recently, different methods and strategies have been reported for fabrication of electrochemical sensing platforms. What are the advantages of this method using electroactive covalent organic framework with the bioactive enzyme?

2.      The traditional single-signal electrochemical sensors have low accuracy and sensitivity and poor reproducibility because their electrochemical signal are easily affected by the background current of the workstation and environmental conditions such as temperature and pH. Please provide the corresponding works! Recent works (e.g., Novel 3D Printed Device for Dual-Signaling Ratiometric Photoelectrochemical Readout of Biomarker Using λ-Exonuclease-Assisted Recycling Amplification; Dual-channel photoelectrochemical ratiometric aptasensor with up-converting nanocrystals using spatial-resolved technique on homemade 3D printed device) can be referred on this topic.

3.      It was known that when at the optimum pH value, the binding ability of the enzyme molecule to the substrate was the strongest and the enzyme reaction rate was the highest, while pH was too large or too small, the enzyme might be inactivated. Relative works (e.g., Novel enzyme-encapsulated DNA hydrogel for highly efficient electrochemical sensing glucose; Near-infrared light-excited core-core-shell UCNP@Au@CdS upconversion nanospheres for ultrasensitive photoelectrochemical enzyme immunoassay) should be mentioned for this description.

4.      The mechanism on the electrochemical sensing platform for the quantitative monitoring of carbaryl should be further discussed and explained in the main text.

5.      How to evaluate the accuracy of electrochemical sensing platform for the monitoring of carbaryl in the real samples? Please simply specify them in the main text.

Author Response

Thank you very much for giving us very valuable suggestions. We have considered your questions seriously, and answered them as below.

Reviewer 2: This manuscript constructed an electrochemical sensing platform for the quantitative monitoring of carbaryl. The ratiometric electrochemical sensor was fabricated by using acetylcholinesterase-loaded electroactive covalent organic framework. The electroactive COF was modified with thionine. Results indicated that the developed electrochemical sensing system could exhibit good analytical properties. However, modification is requested for the further consideration as follows:

1.Q: Recently, different methods and strategies have been reported for fabrication of electrochemical sensing platforms. What are the advantages of this method using electroactive covalent organic framework with the bioactive enzyme?

R: Thank you very much for your relevant advice. Firstly, positively charged COFThi−TFPB was conducive to maintaining the biological activity of AChE and achieving the best catalytic effect. Furthermore, its inherent redox peak at 0/-0.22 V was inert to the detection of carbaryl, which could be used as a reference signal to further improve the sensitivity and accuracy of the detection. Secondly, as previously reported, AChE could selectively catalyze the hydrolysis of ATCh to form TCh, and its catalytic activity could be specifically inhibited by carbamate pesticides, making it a target substance for monitoring carbamate pesticides. In this paper, the appropriate support material COFThi-TFPB was used to immobilize AChE to construct a biological interface that was conducive to electron transport, and the content of pesticide carbaryl was quantitatively detected by AChE inhibition-based electrochemical method. The corresponding discussion has been added in the revised manuscript. (See the revised manuscript)

2.Q: The traditional single-signal electrochemical sensors have low accuracy and sensitivity and poor reproducibility because their electrochemical signal are easily affected by the background current of the workstation and environmental conditions such as temperature and pH. Please provide the corresponding works! Recent works (e.g. Novel 3D Printed Device for Dual-Signaling Ratiometric Photoelectrochemical Readout of Biomarker Using λ-Exonuclease-Assisted Recycling Amplification; Dual-channel photoelectrochemical ratiometric aptasensor with up-converting nanocrystals using spatial-resolved technique on homemade 3D printed device) can be referred on this topic.

R: Thank you very much for your relevant advice. According to your suggestion, relevant literatures have been cited as Ref 42 and 43 in the revised manuscript. (See line 12-17 on page 18 in the revised manuscript).

3.Q: It was known that when at the optimum pH value, the binding ability of the enzyme molecule to the substrate was the strongest and the enzyme reaction rate was the highest, while pH was too large or too small, the enzyme might be inactivated. Relative works (e.g.,Novel enzyme-encapsulated DNA hydrogel for highly efficient electrochemical sensing glucose; Near-infrared light-excited core-core-shell UCNP@Au@CdS upconversion nanospheres for ultrasensitive photoelectrochemical enzyme immunoassay) should be mentioned for this description.

R: Thank you very much for your relevant advice. According to your suggestion, relevant literatures have been cited as Ref 59 and Ref 60 in the revised manuscript. (See line 20-23 on page 20 and line 1 on page 21 in the revised manuscript).

4.Q: The mechanism on the electrochemical sensing platform for the quantitative monitoring of carbaryl should be further discussed and explained in the main text.

R: Thank you very much for your relevant advice. According to your advice, the mechanism for quantitative monitoring of carbaryl have been revised as followed. “The mechanism of the pesticide electrochemical sensor is that the carbamate pesticides will make the serine residue hydroxyl group of AChE catalytic center be carbamylated, resulting in a decrease in its activity, or even completely inactive. So the catalytic hydrolysis of ATCh is weakened, leading a decrease in the production of thiocholine (TCh). The response current of TCh is inversely proportional to the concentration of carbamate pesticides such as carbaryl, so as to realize the simple, efficient and sensitive detection of carbaryl [27-31]” in the revised manuscript. (See line 23 on page 3 and line 1-6 on page 4 in the revised manuscript).

5.Q: How to evaluate the accuracy of electrochemical sensing platform for the monitoring of carbaryl in the real samples? Please simply specify them in the main text.

R: Thank you very much for your relevant advice. According to your advice, the real examples containing carbaryl have been checked by using AChE/COFThi-TFPB/GCE sensor and high performance liquid chromatography (HPLC) to demonstrate the application potential of sensor in Table 2 (Table R2). As shown in Table R2, the carbaryl content in the actual samples obtained by AChE/COFThi-TFPB/GCE sensor was close to the results of the HPLC test, which proved that the AChE/COFThi-TFPB/GCE sensor has the potential to detect carbaryl in real examples. The corresponding discussion has been added in the revised manuscript. (See line 3-11 on page 11 in the revised manuscript). The modified Table 2 is shown in Table R2 (See line 2 on page 34 in the revised manuscript).

Round 2

Reviewer 1 Report

The manuscript has been improved for publication in Biosensors.